Arbuscular mycorrhizal fungi enhanced the growth, phosphorus uptake and Pht expression of olive (Olea europaea L.) plantlets

Wu Tao 1
Pan Li 1
Zipori Isaac 2
Mao Jihua 1
Li Rongbo 1
Li Yongpeng 1
Li Yongjie 1
Jing Yuebo 1401764297@qq.com 1
Chen Haiyun kmchenhy@163.com 1
1 Yunnan Academy of Forestry and Grassland , Kunming , Yunnan , China
2 Gilat Research Center, Agricultural Research Organization , Negev , Gilat , Israel
Wang Xiukang
Electronic publication date: 2022 Aug 9
Publication date: 2022
Volume: 10
Electronic Location ID: e13813
Received 2022 Feb 21; Accepted 2022 Jul 8
Copyright: ©2022 Wu et al.
Copyright year: 2022
Copyright holder: Wu et al.
License: This is an open access article distributed under the terms of the Creative Commons Attribution License, which permits unrestricted use, distribution, reproduction and adaptation in any medium and for any purpose provided that it is properly attributed. For attribution, the original author(s), title, publication source (PeerJ) and either DOI or URL of the article must be cited.
License URL: https://creativecommons.org/licenses/by/4.0/

Keywords: Olea europaea L, Native arbuscular mycorrhizal fungi, Growth promotion, Phosphorus uptake, Phosphate transporter gene

Funding: The National Natural Science Foundation of China No. 31760222 The “Plant Nutrition and Mycorrhiza Research” Discipline Team Building Project of Yunnan Academy of Forestry and Grassland No. LKYTD-2018-6 This study was funded by the National Natural Science Foundation of China (No. 31760222), and the “Plant Nutrition and Mycorrhiza Research” Discipline Team Building Project of Yunnan Academy of Forestry and Grassland (No. LKYTD-2018-6). The funders had no role in study design, data collection and analysis, decision to publish, or preparation of the manuscript.

==============================
Olive (Olea europaea L.) is a highly mycotrophic species that has been introduced and cultivated in China for half a century. The arbuscular mycorrhizal fungi (AMF) is extremely valuable as a kind of biofertilizer to promote the health and vigor of olive plants. However, it is still unclear how native AMF impact growth and mineral nutrients, especially phosphorus absorption in the area where olive trees were introduced in China. In the present study, through a pot experiment, the effects of native AMF on the growth, phosphorus uptake and expression levels of four phosphate transporter genes (Pht) of olive plantlets were characterized. We found that (1) typical AMF colonization was observed within the roots of inoculated olive plantlets, and the growth of plantlets was significantly promoted; (2) some indigenous consortia (AMF1 and AMF2) notably promoted the absorption of phosphorus, fertilizers significantly increased the foliar content of nitrogen, and both AMF inoculation and fertilization had no significant effect on the uptake of potassium; and (3) AMF inoculation enhanced the expression of phosphate transporter genes in inoculated olive roots. This work demonstrates the effectiveness of native AMF on the cultivation of robust olive plantlets and highlights the role of AMF in increasing phosphorus uptake. There is great potential in using native AMF consortia as inoculants for the production of healthy and robust olive plantlets.

Introduction

Olive (Olea europaea L.), a multifunctional long-living tree crop, is relevant not only for table olive and oil production but also for its impact on human nutrition and rural lifestyle (Fabbri, Lambardi & Ozden-Tokatli, 2009). Olive has long been the symbol of the Mediterranean and is now spreading in many other new areas (Gutierrez, Ponti & Cossu, 2009; Horden & Purcell, 2012). A global production trend of olive has been on the rise and the countries that were previously importers of the product, such as the US, China, Chile and Australia are now producing. Olive trees were introduced to China in the 1960s, and now grown in 14 provinces, mainly Gansu, Sichuan, and Yunnan, covering an area of 167,000 ha (Wang et al., 2019). These areas are located in western China, and the soil is mostly acidic red soil and yellow soil, phosphorus (P) is one of the major plant nutrients that is least available in the soil. Aluminum and iron ions, which predominate in acidic soils, interact strongly with P and render it unavailable to plants (Raghothama & Karthikeyan, 2005).

Arbuscular mycorrhizal fungi (AMF) can form mutualistic symbioses with approximately 80% of land plant species. Notably, AMF are known to improve phosphorus (P) nutrition in plants by making them accessible to unavailable soil P sources (Allen, 1996). They can acquire nutrients from soil volumes that are inaccessible to roots, and provide the host plant with mineral nutrients and water, in exchange for photosynthetic products (Berruti et al., 2016). Olive is a typical mycotrophic species (Calvente et al., 2004; Hayman, Barea & Azcon, 1976; Roldán-Fajardo & Barea, 1986), and many studies have reported that the early presence of AMF increases the growth of olive rooted cuttings and micropropagated plants (Estaún et al., 2003; Martín et al., 2006) and enhances olive plant growth (Aganchich et al., 2022), absorption of nutrients (Rodrigues et al., 2021), physiological parameters (Tekaya et al., 2017), tolerance to stress caused by transplanting (Bompadre et al., 2014; Dag et al., 2009), drought (Ouledali et al., 2018), salinity (Ben Hassena et al., 2021) and disease (Boutaj et al., 2021). Thus, AMF show superior prospects as biofertilizers, especially in tropical soils (Igiehon & Babalola, 2017), which are usually dominated by iron and aluminum oxides and maintain a lower available P than temperate soils (Tiessen, 2005).

The content of P in plant tissue is 500-2000 times that of soil available phosphorus (Bieleski, 1973; Raghothama & Karthikeyan, 2005). The ability of AMF or bioinoculants to assimilate P is one of the key factors for plants to effectively utilize soil P. The high expression of the plant’s own phosphorus transporter gene (Pht) is another important factor (Nussaume et al., 2011). This sharp concentration gradient between plants and soil illustrates the critical role of Pht genes. A majority of Pht genes exhibit strong expression in roots, a few Pht genes express in aerial organs such as leaves or flowers (Nussaume et al., 2011). After the plant roots were induced by AMF, the expression levels of Pht genes increased sharply (Nagy et al., 2005; Xu et al., 2007; Yang et al., 2012). However, the specific situations in olives are not clear.

In this study, we hypothesized that, after years of planting in China: (1) the olive trees harbored some indigenous AMF; (2) these AMF can produce promising effects on the growth and phosphorus (P) uptake of olive plants, and the expression of genes related to P uptake can also be promoted with the root colonization of AMF. In this context, we collected rhizosphere soil of olive trees from five growing sites in Yunnan as the source of indigenous AMF consortia and compared them with commercial AMF inoculum and fertilizer to study their effects on olive plantlets. Furthermore, the gene expression of P absorption via the formation of AM was also characterized.

Materials & Methods

Plant materials and cultivation conditions

Olive cultivar ‘Koroneiki’ was selected since it is one of the most common olive cultivars in Yunnan province with promising performance (Li et al., 2020). The plantlets used in this study were obtained by mist propagation. The semi-woody cuttings of approximately 10 cm long and two pairs of leaves at the top end were selected for the study. The bottom basal end of each cutting was immersed in an ethanol solution containing 5 g of indol-3-butyric acid/L to promote root development. Propagation was performed in an arch tunnel with a controlled glass greenhouse under an air temperature of 20 ± 2 °C, a relative humidity of 55–70%, a range of irradiance of 400–650 µmol photon m−2 s−1 (Martín et al., 2006). After three months, the rooted cuttings were transferred to 7.5 cm × 11.5 cm plastic pots filled with growing medium.

The growing medium was a mixture of 50% peat moss and 50% vermiculite (v/v) with a pH of 6.85, organic matter of 251.83 g/kg, hydrolyzable nitrogen of 406.76 mg/kg, available phosphorus of 43.12 mg/kg, available potassium of 469.38 mg/kg, total nitrogen, phosphorus and potassium of 5.08 g/kg, 0.62 g/kg and 16.57 g/kg, respectively. The growing medium was autoclaved twice for a period of 2 h at 121 °C, with a 24 h interval between the two sterilizations.

AMF and fertilization treatments

The native soil inocula were collected from five orchards located in the representative olive introduction and cultivation areas in Yunnan Province. The five growing sites sampled were coded AMF1∼AMF 5. AMF1 was sampled from the orchard of Olive Cultivation Technology Extension Base of Diqing Forest Institute of Yunnan Province. AMF2 from the orchard of Yunnan Lvyuan Industrial Development Company Limited. AMF3 from the orchard of Chuxiong Xinyuan Biotechnology Company Limited. AMF4 from the orchard of Lijiang Tianyuan Olive Technology Development Company Limited. AMF5 from the orchard of Lijiang Sanquan Olive Industrial Development Company Limited. For specific location information, please refer to Table S1. The spore numbers of AMF1 to AMF5 were 350, 121, 82, 198 and 116 in 20 grams of soil. The dominant AMF species of AMF1, AMF2, AMF3 and AMF5 were Funneliformis geosporum, that of AMF4 was Septoglomus constrictum (see the Table S1), the detailed AMF communities characterization was reported in our previous work (Jing, Mao & Li, 2022). A commercial AMF inoculant (coded AMF6) was assayed as a reference alongside the native soil inocula. AMF6 was produced in the form of granules, spores and roots, viz. Rhizophagus intraradices, F. mosseae, R. aggregatus and Claroideoglomus etunicatum (without other additives). Three fertilizer treatments were used here to compare their effects with AMF inocula, namely, Fert1 (3 gram/pot of compound (15:15:15) fertilizer), Fert2 (3 gram/pot of earthworm manure, whose main components include 56.2% organic matter, 1.78% total nitrogen, 1.67% total phosphorus, 1.02% total potassium, pH 7.66) and Fert3 (Fert1 + Fert2). Plantlets with neither inoculation nor fertilizer application were used as the control (CK). Six repetitions were performed for each treatment. Inoculation was carried out at the time of transferring the rooted cuttings into the containers. Three grams of inoculum was deposited directly below the roots of the rooted cuttings.

To expose the cuttings in every treatment to an equal nutritional level, for the AMF1 treatment, 3 g of sterilized rhizosphere soil from AMF2 to AMF5 were added to the pot, and for AMF2, 3 g of sterilized rhizosphere soil from AMF1, AMF3 to AMF5 were added to the pot, and so on. For the AMF6, Fert1, Fert2, Fert3 and CK treatments, the rhizosphere sterilized soil of AMF1 to AMF5 was added to the pot, at the rate of 3 g each.

The plantlets were grown in the greenhouse of the Yunnan Academy of Forestry and Grassland and watered manually with tap water The frequency of watering depended on the environmental conditions prevailing inside the greenhouse, generally once a week. After six months of growth, the shoots and roots of the olive plantlets were harvested separately.

Detection of AMF colonization

The roots were first washed with tap water, and randomly collected root segments were then cleared with 10% (w/v) KOH at 90 °C in a water bath for approximately 60 min. After cooling to room temperature, root samples were thoroughly washed with tap water, stained with blue ink (Hero® 203; Hero, Shanghai, China), mounted on microscope slides and then examined under a compound light microscope (Olympus-BX53; Olympus Corporation, Tokyo, Japan) for the presence of AM fungal structures. The percentage of root length occupied by hyphae, arbuscules and vesicles was quantified on each sample by a modified line intersection method (McGonigle et al., 1990). At least 200 intersections per root sample were examined. Four replicates were performed for each treatment and control.

Plantlet growth measurements

Growth parameters were measured for all plantlets, including height, root collar diameter, and above- and belowground biomass. The dry weights of the shoots and roots were recorded after oven-drying at 70 °C until they reached a constant mass. The foliar nutrient concentration was determined on dried material in the tested plantlets. Dried leaves were milled and passed through a 0.25 mm sieve, and a sample of 0.5 g of leaf powder was taken for digestion with H2SO4 and H2O2 (v/v) 1:4. Nitrogen, P and K contents were determined by the Kjeldahl Method (Foss Kjeltec 8400; Foss Analytics, Hillerød, Hovedstaden, Denmark), ultraviolet–visible spectrophotometry (Hitachi U-5100; Hitachi, Tokyo, Japan) and atomic absorption spectrophotometry (Hitachi Polarized Zeeman Atomic Absorption Spectrophotometer ZA3000; Hitachi, Tokyo, Japan), respectively. In addition, the ratio of root:shoot growth was calculated to reflect the robustness of the plantlets and the efficiency of AMF inoculation (Tobar, Azcón & Barea, 1994). Six replicates were performed for each treatment and control.

qRT–PCR of phosphate transporter genes

The wild olive assembly and annotated genome (Unver et al., 2017) were used as queries for phosphate transporter protein genes (Pht). Firstly, we use “phosphate transporter” as the keyword to search target genes in the wild olive genome of Olea europaea var. sylvestris (RefSeq assembly accession: GCF_002742605.1), then obtained the name and location information of target genes, found the corresponding protein sequence. Secondly, we got 30 gene members of the three phosphate transporter gene families based on the verifying of sequence length and function annotation information. Thirdly, we randomly selected several genes from each gene family as representatives, synthesized primers and amplified them. Finally, four olive Pht genes with good effect were selected to analyze their gene expression level in this study. According to the annotation and retrieval results of the wild olive genome, they are named OePht1;11, OePht1;2, OePht1;4 and OePho3 respectively. The four olive Pht studied here refer to Table S2.

To understand the homology of the four olive Pht genes with other known crop ones, a phylogenetic tree was constructed using their DNA sequences together with 11 known Pht genes ones in other crops, MtPHT2;1, MtPT1 and MtPT4 from alfalfa [Medicago truncatula] (Harrison, Dewbre & Liu, 2002; Versaw & Harrison, 2002), LePT3 and LePT4 from tomato [Lycopersicon esculentum] (Xu et al., 2007), StPT1, StPT3 and StPT4 from potato [Solanum tuberosum] (Nagy et al., 2005), OsPT11 and OsPT13 from rice [Oryza sativa] (Paszkowski et al., 2002; Yang et al., 2012) and ZmPT6 from maize [Zea mays] (Wright et al., 2005). All 15 nucleotide sequences representing 6 species were aligned in MAFFT program, then the phylogenetic tree was constructed with MrBayes under the General Time Reversible (GTR) nucleotide substitution model and 10,000 burn-in length in Geneious Prime 2020.0.5 software.

Total RNA of olive roots was isolated using an RNAprep Pure Plant Kit (Tiangen, China). DNase was used to eliminate the potential trace of genomic DNA in RNA samples. Then, 1.0% agarose gel and a NanoDrop ND-2000 spectrophotometer (Thermo Fisher, USA) were used to evaluate and quantify RNA, respectively. RNA samples were reverse-transcribed into cDNA with the FastKing RT Kit (Tiangen, Beijing, China), and synthesized cDNAs were used as templates for qRT–PCR with the SuperReal PreMix Plus Kit (Tiangen, China). O. europaea translation elongation factor-1 alpha (OeEF1α) served as the internal control for the qRT–PCR (Ray & Johnson, 2014). All the primer sequences are listed in Table 1.

Table 1 Primer sequences of four Pht genes and the internal control gene OeEF1α in the root of Olea europaea.

Gene name	Accession number	Sequences (5′–3′)	Produce size/bp	
Pht1;11	NC_036237	F–ATCCACTTGCCACTCACTGA
R–ATATCTCCTCCAGCGACAGC	201	
Pht1;4	XM_023017225	F–GACTGCGATCTACATGCCATG
R–GCCTAACACGATGAGCGAATT	164	
Pht1;2	XM_023031284	F–GCTCAAGAATCAACGAGGTCA
R–CGAGTTGGCTGAGACGCATTA	159	
Pho3	XM_023005125	F–AGCACATATTGGGACATTGTA
R–CAGGCTAACCTTAACAAGACA	143	
OeEF1α	XM_002527974	F–GAATGGTGATGCTGGTTTCG
R–CCCTTCTTGGCAGCAGACTTG	191	

A 20 µL reaction solution containing 1 µL cDNA (20 ng), 1 µL of each primer (10 µM), 10 µL SYBR Green I Master mix reagent (Tiangen) and 7 µL ddH2O was amplified with a LightCycler96 (Roche, Basel, Switzerland). The qRT–PCR program was as follows: 95 °C for 15 min, followed by 45 cycles at 95 °C for 10 s, 55 °C for 10 s and 72 °C for 10 s. The qRT–PCR was performed with three biological replicates, and the data are shown as the mean ± SD. The relative transcript level was calculated using the 2-ΔΔCt method (Ray & Johnson, 2014). Three biological replicates were performed for each treatment and control.

Statistical analysis

Values of mycorrhizal parameters were summarized for each native and commercial AMF using some descriptive statistics viz. mean, standard error (SE). Data related to olive plantlet growth parameters per treatment were visualized using boxplots. Differences in AMF colonization intensities, plant growth parameters, the leaf concentrations of N, P and K between treatments, the expression level of Pht genes, were analyzed using one-way analysis of variance (ANOVA), comparisons among mean values in each treatment were made using the Tukey’s test (HSD) (p < 0.05). The significance level adopted for all statistical tests was of 5% and the SPSS Statistics 24 (SPSS Inc., Chicago, IL, USA) was utilized for the statistical analysis. Boxplot diagrams were drawn using the R package (R Core Team, 2019). All values were reported as means and standard errors.

Results

Root colonization by AMF

No mycorrhizal colonization was detected, as expected, in the roots of plantlets under fertilizer application (treatment Fert1, Fert2 and Fert3) and noninoculated ones (treatment CK). Except for AMF4, all other treatments showed similar or higher root AMF colonization percentages compared with the commercial AMF inoculum (AMF6). The percentage of AMF hypha colonization in roots of olive plantlets ranged from 40.20% in AMF4 to 93.78% in AMF3, 3.32% (AMF4) to 42.46% (AMF5) for the arbuscule colonization, 2.07% (AMF4) to 49.56% (AMF5) for the abundance of vesicles (Table 2). Some typical structures of AMF colonizing the roots of olive plantlets are presented in Fig. 1.

Table 2 Root colonization parameters of olive plantlets inoculated with six AMF inocula.

Values are means ± SE (n = 4), columns marked with different letters differed significantly (p < 0.05).

Treatment	AMF hypha colonization (H%)	AMF arbuscule colonization (A%)	AMF vesicle colonization (V%)	
AMF1	73.78 ± 2.35b	17.07 ± 4.40c	12.11 ± 3.14b	
AMF2	78.30 ± 9.76ab	20.80 ± 2.84c	10.91 ± 1.88b	
AMF3	93.78 ± 2.13a	37.69 ± 5.23ab	47.35 ± 8.17a	
AMF4	40.20 ± 6.51c	3.32 ± 0.53d	2.07 ± 1.05c	
AMF5	93.38 ± 4.19a	42.46 ± 7.62a	49.56 ± 11.95a	
AMF6	67.17 ± 5.23b	28.79 ± 4.18bc	11.53 ± 2.92b	

Figure 1 Typical structures of AMF colonizing the roots of olive plantlets.

Plantlets inoculated with AMF1 (A), AMF2 (B and C), AMF3 (D), AMF4 (E), AMF5 (F), and AMF6 (G). Plantlets with compound fertilize application, the treatment Fert1 (H). Uninoculated plantlets, the treatment Control (I). H: hypha, HC: hyphal coil, A: arbuscule, V: vesicle.

Effect of AMF inoculation on the growth of olive plantlets

Six months after inoculation, all AMF inoculation treatments had a positive influence on plant growth in terms of height, diameter, biomass and leaf area (Fig. 2). Compared to the noninoculated CK, the commercial AMF inoculum increased the shoot and root fresh weights of plantlets by 66.27% and 91.90%, respectively. However, the native AMF inocula achieved higher effects (Fig. 2).

Figure 2 Box plots representing growth parameters of six-month-inoculated plantlets by native and commercial AMF, fertilizers, control (uninoculated).

(A) Plant height; (B) Root collar diameter; (C) Shoot fresh weight; (D) Shoot dry weight; (E) Root fresh weight; (F) Root dry weight; (G) Leaf area; (H) Root:shoot ratio. Solid white circles indicate the means, bold black line indicates the median, whereas black dots are outliers. Letters on white circles are the results of Tukey’s HSD tests, where different letters indicate significant differences at P ≤ 0.05.

AMF3 showed the highest effect on aboveground growth, and the shoot dry weights of inoculated plantlets were 2.16 times, 1.41 times, 2.02 times, 1.38 times and 1.24 times those of the CK, Fert1, Fert2, Fert3 and AMF6 treatments, respectively. Whereas AMF1 showed the highest effect on enhancing the belowground growth of the plantlets, the plantlets inoculated with AMF1 had root dry weights 2.84 times, 2.17 times, 2.71 times, 2.19 times and 1.35 times those of treatment CK, Fert1, Fert2, Fert3 and AMF6 treatments, respectively. The root to shoot ratios of olive plantlets inoculated with AMF1, AMF2, AMF3, AMF4, AMF5, AMF6, Fert1, Fert2, Fert3 and CK were 0.64, 0.52, 0.33, 0.48, 0.49, 0.44, 0.34, 0.35, 0.37 and 0.39, respectively.

Effect of AMF inoculation on the N, P, and K contents in olive plantlets

The leaf nitrogen (N) contents of all the AMF inoculated plantlets were lower than the CK, except AMF2, which had a similar value to the control. Fertilization significantly increased the foliar N content compared with AMF and CK (Fig. 3A). The foliar P contents of olive plantlets inoculated with AMF1 and AMF2 were significantly higher than those of all other treatments (Fig. 3B). AMF inoculation marginally enhanced the potassium (K) uptake of the plantlets, and the leaf K contents of plantlets treated with AMF1 to AMF6 were 8.34%, 15.54%, 9.33%, 12.07%, 9.20% and 16.29% higher than those of the control; however, the differences were not significant (Fig. 3C).

Figure 3 Box plots representing the foliar nutrient content of olive plantlets treated with AMF, fertilizers and the control.

(A) Nitrogen content (N content); (B) Phosphorus content (P content); (C) Potassium content (K content). Labeling is same as in Fig. 2.

Effect of AMF inoculation on expression levels of phosphate transporter genes

A phylogenetic tree was constructed using a multiple DNA sequence alignment of four olive Pht genes and ones of other plants (Fig. 4). The four olive Pht genes were clustered into three main groups, OePht1;11 and OePht1;2 were clustered together with PT4 genes of tomato (LePT4), potato (StPT4) and alfalfa (MtPT4); OePho3 and OePht1;4 were individually clustered into a separate group.

Figure 4 Phylogenetic relationships of the Pht DNA sequences in different plants built using MrBayes method based on the GTR substitution model, supported by bootstrap resampling with 1,000 replications.

Le: Lycopersicon esculentum; Mt: Medicago truncatula; Oe: Olea europaea; Os: Oryza sativa; St: Solanum tuberosum; Zm: Zea mays.

We characterized the expression level of the four Pht genes in olive roots inoculated with AMF or fertilized with compound fertilizer and earthworm manure (Fig. 5). One member, Pht1;11, exhibited strong expression in olive roots inoculated with AMF, especially in the plants treated with AMF6, which of expression level jumped as much as 3400-fold compared to the noninoculated control. Pht1;4 was only expressed at high levels in the AMF1 treatment. The low levels of Pht1;2 transcripts in both the AMF and fertilizer treatments were completely different from those in the control. The expression level of the Pho3 gene responded positively to inoculation and fertilization, but fertilization seemed more effective.

Figure 5 Expression levels of Pht1;11, Pht1;4, Pht1;2 and Pho3 in olive mycorrhizal roots by qRT-PCR analysis.

Discussion

Our results showed that the presence of AMF native to the olive growing sites of Yunnan Province significantly promoted the growth, biomass and P uptake of olive plantlets. Several earlier studies have shown that AMF can promote olive plantlet growth and nutrient uptake (Calvente et al., 2004; M’barki et al., 2018; Porras-Soriano et al., 2009). In addition, we noted a distinct mycorrhizal compatibility among the native soil inoculum indicated by differences in root colonization intensity and the effectiveness on plantlets (Table 2, Fig. 2). In fact, different growth responses of olive plants have been demonstrated following inoculation with different AMF strains (Calvente et al., 2004; Castillo et al., 2006; Meddad-Hamza et al., 2010). Moreover, the effects of colonization by AMF on olive plant growth also varied with the plant cultivars. Dag et al. found that the response intensity in terms of height and biomass production of 12 commercial olive cultivars, inoculated with G. mosseae and G. intraradices was highly cultivar specific (Dag et al., 2009). Other researchers (Fouad et al., 2014; Martín et al., 2006) reported that specific compatibility relationships may exist among symbionts, and underscore the importance of host-AMF selection to maximize olive performance.

In this study, we found that the effects of native AMF were generally higher than those of the commercial AMF inoculum (Fig. 2). Many studies have pointed out the higher efficiency of native AMF compared to nonnative, introduced AMF (Affokpon et al., 2011; Briccoli Bati, Santilli & Lombardo, 2015; Estrada et al., 2013; Labidi et al., 2015). Our results are consistent with those found by Chenchouni, Mekahlia & Beddiar (2020), demonstrating that the effects of local AMF strains on increasing different growth parameters of olive plantlets were better than those of commercial AMF species (Chenchouni, Mekahlia & Beddiar, 2020). Several studies have highlighted that different isolates within the same species, rather than different species, can cause large variations in plant response (Angelard et al., 2010; Gai et al., 2006; Munkvold et al., 2004). A study conducted earlier by Calvente et al. (2004), showed that the G. intraradices strain, isolated from the olive rhizosphere, was more effective than the exotic G. intraradices from the culture collection. More importantly, exploring and exploiting native AMF can avoid any potential problems related to the application of nonnative AMF inoculum in terms of biodiversity losses and homogenization as a result of anthropogenic translocation of biota between biogeographic regions (Pellegrino et al., 2012; Schwartz et al., 2006).

There is evidence that AMF play a role in the uptake of nitrate and ammonium which are assimilated and transported within the mycelium as arginine (Olsson, Burleigh & Van Aarle, 2005), but compared with ectomycorrhizas, rates of N uptake by the hyphae of AMF are too small to contribute substantially to the N nutrition of plants (Smith & Read, 2008). Accordingly, N uptake was not significantly different in AMF inoculated plants compared to noninoculated plants in this study (Fig. 3). The lower N concentration of inoculated olive plantlets can be explained by a ‘dilution’ effect commonly observed in plants growing well in infertile conditions (Steenbjerg & Jakobsen, 1963), as Dela Cruz et al. (1988) reported previously in Albizia falcataria seedlings (Dela Cruz et al., 1988). In the present study, the N concentrations of olive plantlets inoculated with AMF1 to AMF6 were 0.8346%, 0.9927%, 0.7827%, 0.7528%, 0.7730% and 0.6939% respectively, whereas the noninoculated poorly growing olive plantlets (CK) showed the higher N concentration (1.0047%) (Figs. 2, 3).

Mycorrhizal symbiosis was found to be important for root system development, which is critical for better mineral nutrition and stress resistance of seedlings. Tobar, Azcón & Barea (1994) conclusively demonstrated that the root-to-shoot ratio reflects the degree of efficiency of AM fungi (Tobar, Azcón & Barea, 1994). In the present study, inoculation with AMF3 significantly enhanced the aerial parts of olive plantlets, resulting in a low ratio of roots to shoots, whereas the plantlets treated with the other four soil inocula achieved root to shoot ratios higher than those of the control. In addition, the average root to shoot ratio of the plantlets treated with fertilizer was the lowest among all the treatments (Fig. 2). Under chemical fertilization, minerals are immediately available for the plant, which reduces the need and triggering for extensive root development, resulting in a low ratio of root to shoot. Later, when the young plants are transplanted in the field without fertilizers, an abrupt decrease in the nutrient uptake and growth rate occurs (Meddad-Hamza et al., 2010).

Many studies have reported that mycorrhizal colonization can enhance P absorption by plants (Abdel-Fattah et al., 2014; Black, Mitchell & Osborne, 2000; Bücking & Shachar-Hill, 2005), including olive plants (Briccoli Bati, Santilli & Lombardo, 2015; Dag et al., 2009; Estaún et al., 2003). Similarly, the results obtained from this study showed that the leaf P content of AMF inoculated plants was higher than that of noninoculated plants (Fig. 3). In addition to dissolving insoluble phosphates into inorganic orthophosphate (Pi) and absorbing Pi from the rhizosphere beyond root depletion zones by AMF mycelium and transport to host plants (Harrison & Van Buuren, 1995), AMF can also induce the expression of some particular phosphate transporter genes in their host plants under Pi-deficient conditions (Rausch et al., 2001). For example, several AM symbiosis-induced Pht genes in plant roots have been identified, such as LePT3 and LePT4 in tomato (Lycopersicon esculentum) (Xu et al., 2007); StPT1, StPT3 and StPT4 in potato (Solanum tuberosum) (Nagy et al., 2005); MtPT1, MtPT4 and PHT2;1 in alfalfa (Medicago truncatula) (Harrison, Dewbre & Liu, 2002; Versaw & Harrison, 2002); OsPT11 and OsPT13 in rice (Oryza sativa) (Paszkowski et al., 2002; Yang et al., 2012); ZmPT6 in corn (Zea mays) (Wright et al., 2005); PtPT8 and PtPT10 in black cottonwood (Populus trichocarpa) (Loth-Pereda et al., 2011). The transcripts of one tomato Pht gene (LePT4) exhibited significantly increased expression levels in low-P treatment and colonized by the Glomus intraradices (Xu et al., 2007). Potato’s Pht genes (StPT4 and StPT5) also exhibited mycorrhiza upregulation when inoculated with Gigaspora margarita, and both were highly functionally redundant (Nagy et al., 2005). In our study, the four olive Pht genes were differentially regulated, with OePht1;11 exhibiting mycorrhiza-specific regulation, similar to LePT4 and StPT4; OePht1;4 only mycorrhiza-upregulated in AMF1; OePht1;2 were not sensitive to AMF and fertilizer; OePho3 seemed more effective to fertilizer than AMF (Fig. 5). Although OePht1;11 and OePht1;2 were closer in the phylogenetic tree (Fig. 4), it remains to be clarified whether they function differently.

Conclusions

Although mycorrhizal networks ubiquitously exist in the soil, in intensively managed fields, mycorrhizal networks are usually absent or low in abundance because of regular soil disturbance destroying the mycelia or the absence of permanent vegetation cover that is needed to maintain mycorrhizal networks. During relatively long periods of development in the nursery, olive plantlets do not have AMF since they are usually grown in inert or fumigated potting media. AMF inoculation at the nursery stages is therefore critical and necessary and can also help plants cope with various stress conditions in the field. In this study, we report the effect of AMF inoculation on growth responses and the expression levels of four phosphate transporter genes in olive plantlets. Research indicates that there is great potential in using native AMF consortia as inoculants for the production of high-quality and robust olive planting stocks, which will help ensure to obtain healthier plantlets being of potentially higher survival rate after transplanting from nursery to the field. Moreover, indigenous AMF are better adapted to local environmental conditions and result in a substantial promotion of plant fitness. The present work concentrated on the response of olive plantlets to AMF inoculation during the nursery process, and further research is required to evaluate the long-term performance of AMF-inoculated plants in the field. In addition, this study also highlights the necessity of further exploring and exploiting the natural diversity of AMF in more olive growing sites and screening for AMF species with a high capacity to alleviate abiotic and biotic stress of olive plants at different growing sites.

Supplemental Information

Supplemental Information 1 The geographic coordinates of the sampling sites, AMF spore contents and dominant species in the soil samples

Click here for additional data file.

Supplemental Information 2 The phosphate transporter gene information in wild olive

Click here for additional data file.

Supplemental Information 3 Raw data

Click here for additional data file.

Additional Information and Declarations

Competing Interests

Author Contributions

Field Study Permissions

Data Availability

The authors declare there are no competing interests.

Tao Wu performed the experiments, analyzed the data, prepared figures and/or tables, authored or reviewed drafts of the article, and approved the final draft.

Li Pan performed the experiments, analyzed the data, prepared figures and/or tables, authored or reviewed drafts of the article, and approved the final draft.

Isaac Zipori conceived and designed the experiments, authored or reviewed drafts of the article, and approved the final draft.

Jihua Mao performed the experiments, analyzed the data, authored or reviewed drafts of the article, and approved the final draft.

Rongbo Li performed the experiments, analyzed the data, authored or reviewed drafts of the article, and approved the final draft.

Yongpeng Li performed the experiments, analyzed the data, authored or reviewed drafts of the article, and approved the final draft.

Yongjie Li performed the experiments, analyzed the data, authored or reviewed drafts of the article, and approved the final draft.

Yuebo Jing conceived and designed the experiments, performed the experiments, analyzed the data, prepared figures and/or tables, authored or reviewed drafts of the article, funding acquisition, and approved the final draft.

Haiyun Chen conceived and designed the experiments, performed the experiments, analyzed the data, prepared figures and/or tables, authored or reviewed drafts of the article, and approved the final draft.

The following information was supplied relating to field study approvals (i.e., approving body and any reference numbers):

The sampling site coded AMF1 was the orchard of Olive Cultivation Technology Extension Base of Diqing Forest Institute of Yunnan Province (Contact person: Mr. Liu Zhongjie, Mobile phone: 13988752457).

The sampling site coded AMF2 was the orchard of Yunnan Lvyuan Industrial Development Company Limited (Contact person: Mr. Mao Zenghui, Mobile phone: 13708480092).

The sampling site coded AMF3 was the orchard of Chuxiong Xinyuan Biotechnology Company Limited (Contact person: Mr. Zu Chaoming, Mobile phone: 13987891588).

The sampling site coded AMF4 was the orchard of Lijiang Tianyuan Olive Technology Development Company Limited (Contact person: Mr. Gao Jinchang, Mobile phone: 15969380365).

The sampling site coded AMF5 was the orchard of Lijiang Sanquan Olive Industrial Development Company Limited (Contact person: Mr. Wang Wenxiong, Mobile phone: 13508888860).

The following information was supplied regarding data availability:

The raw data are available in the Supplemental Files.

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
