# Peer review of "Arbuscular mycorrhizal fungi enhanced the growth, phosphorus uptake and Pht expression of olive (Olea europaea L.) plantlets"

_PeerJ, doi:10.7717/peerj.13813_

## Round 0.1 · original submission · Major Revisions

The paper has research value and significance, and it is suggested to modify it according to the requirements of journal format and reference.

Reviewer 4 has requested that you cite specific references. You may add them if you believe they are especially relevant. However, I do not expect you to include these citations, and if you do not include them, this will not influence my decision.

·

Basic reporting

no comment

Experimental design

no comment

Validity of the findings

no comment

Additional comments

minor questions:
1, the author mentioned the boron in the abstract and results, but the material and methods were not mentioned boron. N, P and K are major elements and boron is minor element, why choose boron as representative of minor element for Oliver ?
2,dose the author measured the N, P, K in the roots and shoot? why just measure the leaf?
3,is there any plant pictures ? it will be more direct evidence for the promotion of plant growth by AM.

Reviewer 2 ·

Basic reporting

Overall, the manuscript is quite well written and structured and as a result I was able to read it very smoothly and quickly. Literature discussed along the text is quite comprehensive and hypotheses were clearly stated. However, I would add to the introduction section a greater focus on the effects of AMF and soil microbiota on olive tree health, as these information were already afforded in the very recent literature.
As a further comment I would recommend authors to better define the rationale of the study concerning some detailed aspects. E.g. for the quantification of Boron: why this element was quantified? It has to be mentioned that Boron is crucial for olive tree flowering and fruit-set stages, for readers who do not known detailed Olive tree physiology.
Moreover, some sections needs to be re-organized. E.g. the paragraph at lines 206-211 should be anticipated as it would be preparatory to the performed expression analyses.
Finally, I would smoothen the first results section in which growth response results were described in too much detail showing the percentage of each ratio.

Experimental design

The experimental design was very clear, and lot of attention was devoted to compensate nutrients level across the treatment, which is really appreciable. However, if most of experimental procedures were described in greater detail, some other fully lacks or further need to be integrated. As an example, the soil sterilization protocol, which is often a critical step, was not provided as well as the geographic coordinates of the sampled soils (I would report them as supplementary materials). Similarly a phylogenetic analysis was reported in Results but methods used to generate alignments and the final tree were not reported at all.
Concerning this point it is not clear how Pt transporter homologs were initially identified (lines 138-139) and further named. At lines 198-205 it would be useful to detail if PT genes numbering was introduced here arbitrarily for olive tree or if it was already present in the genome annotation.
I saw that a pylogenetic tree with the putative PT transporter genes was performed. However, it is not clear if orthologies were tested (e.g. by specialized software which compares proteomes, or at least by the ‘best reciprocal hit’ method). For the sake of clarity I would put in brackets, at the first gene mention, the corresponding homolog genes of a mycorrhiza Eudicots model species such as Medicago or tomato.
More in general, methods should be better referenced, e.g. at line 152 (qPCR analysis methods), and significance threshold levels adopted in statistical tests should be stated (e.g. lines 156-159).

Validity of the findings

Unfortunately, I have not found anywhere the number of biological replications used in the study. It would be crucial to assess the strength of results showed. Moreover, in gene expression analysis differences among means should be tested using a proper statistical posthoc test after ANOVA(in Figure 3 no differences among conditions were reported).Beside these two aspects I found results solid and the performed analysis convincingly sound.
As a conclusion I found that these results well-supports what is known from literature about olive tree and AM symbiosis. For this reason I would further stress in conclusion section the applicative framework of the research, e.g. to obtain healthier plantlets which potentially hold higher survival rate after transplanting from nurseries to the field.

Additional comments

Detailed points:
- line 219: please define how do you mean with “effectivenenss of plantlets”.
- lines 231-232: here it is not clear if the authors means “inconsistent” or the opposite..
- lines 276-277: please update tomato scientific binomial to the currently accepted name Solanum lycopersicum.

·

Basic reporting

Reviewer’s comments
BASIC REPORTING
• Clear, unambiguous, professional English language used throughout. [Yes, it is ok]
• Intro & background to show context. Literature well referenced & relevant. [ Fairly good]
• Structure conforms to PeerJ standards, discipline norm, or improved for clarity. [Ok}
• Figures are relevant, high quality, well labelled & described. [ok]
• Raw data supplied (see PeerJ policy). [Supplied & checked]

EXPERIMENTAL DESIGN
• Original primary research within Scope of the journal. [Yes]
• Research question well defined, relevant & meaningful. It is stated how the research fills an identified knowledge gap. [Some limitations are there which have been mentioned in the reviewed version]
• Rigorous investigation performed to a high technical & ethical standard. [Fairly good]
• Methods described with sufficient detail & information to replicate. [Needs improvement]

VALIDITY OF THE FINDINGS
Impact and novelty not assessed. Meaningful replication encouraged where rationale & benefit to literature is clearly stated. [To be fine-tuned]
All underlying data have been provided; they are robust, statistically sound, & controlled. [these are ok]

Experimental design

Ok.
Some suggestion made in the M&M.

Validity of the findings

Fairly good, however, some suggestion have been made which needs to be done.
Results:
1. Nicely written based on the hypothesis and raw data also provided.
Discussion:
Discussion part is ok, however, I didn't find the requirement of study of B in the growth performance of plantlets and significance of the B in the study. Please justify.

Suggestion to add: Future scope of research

Additional comments

• The article is good and having relevance.
• It needs revision in the direction as suggested.
• Can be considered for publication after revision.

Reviewer 4 ·

Basic reporting

The topic of this experimental article is well-chosen, timely and within the scope of PeerJ. However, Manuscrit needs revision before it is accepted for publication in PeerJ.
The information presented is indeed not complete and needs to be updated. More attention should be paid to the following points:
- The current version lacks a critical evaluation of the results achieved and a treatment of the already known literature, on the subject. A good discussion should not only report what has been written in the past but should critically evaluate the results presented, especially where there are conflicting reports.

The organization of the manuscript is somewhat cluttered and numbering would be helpful. The review should be more concise and the synthesis more elaborate.
I recommend that the Figs be cleaned up and redesigned - they often lack post-hoc tests and clear significance - e.g. Fig4. In addition, some are in colour and others in black and white. All in colour or convert to clear tabs as well.
References should be double-checked and formatting should be done strictly according to journal rules. Authors constantly alternate between citing titles in upper and lower case.
Furthermore, typos were consistently found in the ms and authors are asked to read their ms carefully before submitting them.
Below, I will address some of the other points I came across while reviewing the ms:

Experimental design

It is not clear when the experiments were carried out?
r. 113 - add country
HSD test is not clear from figs.
Figure captions must be self-supporting.

Validity of the findings

Wrong statistics, homogennous group etc. - needs clarification. Experimental design is missing. No clear schema of experiment, Pht1; Pht 11? ? AMF 6? Fert3?

Additional comments

My suggestion regarding this MS is to accept after revising with the following points:
1. Authors presented a good introduction, but they could explain more about molecular mechanisms of plant phosphorus tolerance and potential mitigation from phosphorus toxicity.
2. Introduction and discussion sections authors could involve also new aspects - hormonal regulation of the phosphorus tolerance (interaction with SA, K etc), including new references.
3. Add more view on photosynthetic parameters/discuss, which kind of parameters are more sensitive and why.
4. Add some recent references in the MS. It is important to discuss the plant regulation mechanisms.
Paper brings many new aspects and the novelty of the paper is OK, but I would like to invite authors to discuss also more eco-physiological aspects using new references: DOI: 10.1016/j.scitotenv.2021.147943; 10.1016/j.scienta.2020.109712; 10.3390/agriculture11030194; 10.1002/jsfa.9196; 10.3390/microorganisms10010051; 10.1016/j.agwat.2020.106635; 10.1016/j.apsoil.2020.103520

Final COMMENTS
- The manuscript is useful and innovative, it contains original data.
- This study presents the relevant matter in more depth than some of the other related publications, therefore, I recommend its publication after MODERATE changes.

·

Basic reporting

The introduction needs more logic. I suggest that you improve the description of the relationship between Olive and P for your study as this was not sufficient to support your story. The reason why you focused on the topic of your manuscript was not well expressed to us. For example, Prof. Gu Feng of CAU is an expert in the study area of AMF and P, his group has some findings on your topic, however, it is lacking in your introduction.

Experimental design

M&M: methods are important for any experiments, it is suggested that supplement some literature of yours or others about M&M, especially of Methods in L75-L115.
L76, the references of 'Koroneiki was selected since it is one of the most common olive cultivars in Yunnan province' should be supplemented.
L131, concentration should be content.
L153, what is the statistical software?

Validity of the findings

Results: actually, I can not find that the results support can the conclusion and the tile of Arbuscular mycorrhizal fungi enhanced phosphorus uptake, which should be given more information.

Additional comments

In figure 4, the same italic style of Pht1;11, Pht1;4, Pht1;2 and Pho3, is suggested to express in the abscissa heading of the figure.

---

## Round 0.2 · Minor Revisions

Please follow the journal format requirements and further check the language.

·

Basic reporting

The manuscript is written and structured well. I didn't find any problems with the English written. However, there is still some results mentioned Boron in the results (lines 226-235). Please check and revise. As the author response, it should be comprehensive described and conducted in another article.

Experimental design

no comment

Validity of the findings

no comment

Additional comments

no comment

Reviewer 2 ·

Basic reporting

This is a revised version of a manuscript I previously reviewed. Compared with the first version I found it largely improved since authors integrated all the comments raised by all the five reviewers. However, I still found many missing information and paragraphs that can be largely improved. My further comments here below refers to the “track-change” *.docx version.

Experimental design

The manuscript should be corrected and integrated according to the following points:

- Please check all scientific names are in italics (see e.g. lines 178, 179).
- information on how protein/nucleotide sequences were aligned, manipulated to perform phylogenetic analysis remains still unknown. There are plenty of methods to do so which can results in profoundly different outputs. I would require authors to add these information to ensure reproducibility of results, including all the parameters set to truncate alignments, selects evolutionary models and perform phylogeny.
- I notice that in M&M section authors detail the composition of AMF communities/inocula used. This is especially relevant to understand differences among treatments. However, it is not clear if these data comes from already published studies or it they newly characterized these material. In this latter case they should provide methods used to do so, otherwise a bibliographic reference is required. If available it would be great if they will be able to provide a detailed AMF communities characterization as supplementary material.
- differently to what they stated in their rebuttal letter, the Boron story is still present in the manuscript (see lines 244 and 31). I do not have any issues against it since in my view also negative data (i.e. no effects on B uptake upon mycorrhizal inoculation) is data and can be relevant and useful to other researchers. However, as I already reported previously, it should be better contextualized to guide the reader who does not known its relevance in olive tree.

Validity of the findings

no comment

---

## Round 0.3 · accepted · Accept

The author carefully revised it and agreed to accept it.